# Using SLISEMAP to interpret physical data

**Lauri Seppäläinen**◉*, **Anton Björklund**◉, **Vitus Besel**◉, **Kai Puolamäki**◉

University of Helsinki, Helsinki, Finland

◉ These authors contributed equally to this work.
* lauri.seppalainen@helsinki.fi

## Abstract

Manifold visualisation techniques are commonly used to visualise high-dimensional datasets in physical sciences. In this paper, we apply a recently introduced manifold visualisation method, SLISEMAP, on datasets from physics and chemistry. SLISEMAP combines manifold visualisation with explainable artificial intelligence. Explainable artificial intelligence investigates the decision processes of black box machine learning models and complex simulators. With SLISEMAP, we find an embedding such that data items with similar local explanations are grouped together. Hence, SLISEMAP gives us an overview of the different behaviours of a black box model, where the patterns in the embedding reflect a target property. In this paper, we show how SLISEMAP can be used and evaluated on physical data and that it is helpful in finding meaningful information on classification and regression models trained on these datasets.

## 1 Introduction

Many real-world datasets, including those in physics, are tabular; rows correspond to data items (points) and columns to features of each data item. One problem when dealing with datasets with many features is making meaningful summaries. In manifold visualisation, the objective is to find an embedding that "compresses" the high-dimensional data into, typically, two dimensions. Manifold visualisation has become a central tool for exploring and understanding complex scientific datasets [1–4].

Concurrently, machine learning is increasingly applied in physical sciences [5]. However, powerful supervised learning models and physics simulations are often "black boxes": it is difficult to understand how the properties of the data influence the predictions. If the aim is to understand the underlying phenomena, using black box models is problematic. Additionally, to trust the model predictions, it is helpful to understand by which criteria they are made [6]. Explainable Artificial Intelligence (XAI) is a branch of machine learning that tries to open up these black boxes by extracting more information that could explain the predictions [7].

SLISEMAP [8] is a recently introduced manifold visualisation method that incorporates local explanations. SLISEMAP produces a visualisation where items with similar explanations form visual patterns, such as clusters. A simple example can be seen in Fig 1 where two local explanations, in the form of linear approximations, are enough to explain the complex function and how the points in the embedding form two clusters corresponding to the two explanations.

**Data Availability Statement:** The datasets and source code used in the paper are available at https://www.edahelsinki.fi/papers/slisemap_phys.

**Funding:** We thank the Research Council of Finland (decisions 346376 (LS, KP), 345704 (KP), 337549 (VB) and 34636 (VB)), and the Doctoral

Programme in Computer Science at University of Helsinki (AB) for funding, and the Finnish Computing Competence Infrastructure (FCCI) for supporting this project with computational resources. Open access funded by Helsinki University Library.

**Competing interests:** The authors have declared that no competing interests exist.

While SLISEMAP is introduced in [8], this paper focuses on demonstrating its application to physical datasets.

When analysing physical data, an analyst usually has two goals: (i) to understand the structure of the high-dimensional data and (ii) to understand how this data relates to a target property, such as a class label or continous value. A model of some kind is often used to elucidate the connection between the features and the target. This paper shows examples of how regression models, such as random forests, can classify particle jets from high-energy physics and how quantum chemical simulations are used to estimate molecular properties. SLISEMAP offers both explanations for the target property in the form of local simple models while also providing a visualisation where data points with similar explanations form patterns.

The objectives of this paper are the demonstration of (i) how SLISEMAP can be applied to describe physical datasets (Sect. 3.2), (ii) how the resulting visualisations can be evaluated (Sect. 3.3), and (iii) that the explanations carry meaningful information for the domain expert (Sect. 4).

## 2 Related work

This section briefly reviews related work on manifold visualisation and explainable artificial intelligence (XAI) and how they are used in science.

Manifold visualisation is commonly applied to explore and understand complex data in many fields of science, from genetics [1, 2] and chemoinformatics [3] to astronomy [4] and linguistics [9]. Manifold visualisation aims to present high-dimensional data as a low-dimensional (usually 2D) embedding while preserving the maximum amount of information based on preset criteria. In science, standard manifold visualisation methods include linear projections [10], such as principal component analysis (PCA) [11], and non-linear methods, such as t-SNE [12], and UMAP [13]. For a survey on manifold visualisation techniques, see, e.g., [14].

These are examples of unsupervised methods; they can be utilized to discover structure in data but cannot offer insight into how the features affect the predictions. As a result, their embeddings for the example in Fig 1 would not show two clusters. There is also a supervised variant of UMAP [13], and several other supervised manifold visualisation techniques have been proposed, such as linear low-rank projections [15].

Results from manifold visualisation methods are often analysed via visual inspection. Hence, we need to verify that the emerging patterns reflect some true quality of the data and are not just artefacts produced by the algorithm in question. A wide range of visualisation evaluation metrics has been proposed, from summary statistics such as stress [16] to visualisations such as Shepard diagrams [17]. However, many evaluation metrics rely on the notion of distance both in the input data and the visualisation. Scientific data is often tabular, and features

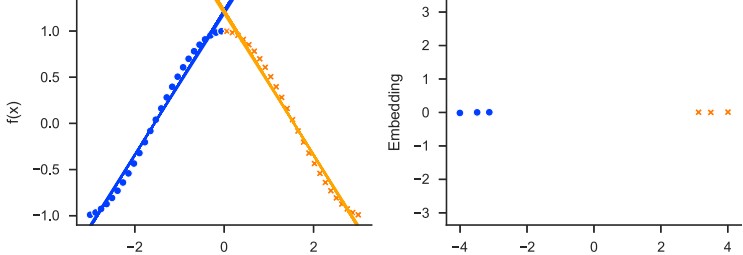

**Fig 1.** Applying SLISEMAP on a dataset with predictions from an unknown function (points in the left plot). SLISEMAP finds local models (lines in the left plot) approximating the complex prediction function for every data item. We also get an embedding (right plot) where data items with similar local models are grouped together.

can combine class variables, vectors and scalar values with different units. Defining suitable distance measures is, therefore, challenging. Furthermore, not all manifold visualisation methods produce linear or even metric embeddings.

Evaluation metrics inspired by information retrieval have been proposed for non-linear embedding methods, such as precision and recall [18]. Incidentally, the definition of precision, which compares neighbourhood relations of the items between the input and embedding spaces, serves as a basis for the neighbourhood quality metric we introduce in Sect. 3.3. For a thorough quantitative survey concerning the evaluation of visualisations, see [19].

Explainable AI seeks to enhance the interpretability of black-box models; see, e.g., [7, 20] for general surveys and [21] for XAI in chemistry. XAI methods can be classified along two axes: *model-specific* vs. *model-agnostic* and *local* vs. *global*. Model-specific methods can only be applied to a single model (class), while model-agnostic methods can be used for any model. Global methods aim to summarise the entire black-box model, while local methods explain limited regions of the data manifold, e.g., near a selected data item.

As black-box models can be very complex, summarising them holistically and interpretably is challenging. As such, many model-agnostic explanation methods provide only local explanations. A common approach is to use an interpretable surrogate model to locally approximate the black-box model [7] in the neighbourhood of the points of interest. A popular choice is to use linear models for the approximation [22–24]. Naturally, for linear models to be interpretable, the number of coefficients cannot be too large [6], and the features in the data also need to be understandable [21, 24].

These methods include LIME [22], SHAP [23] and SLISE [24]. LIME provides sparse linear models as explanations for classifiers, with a penalty function for local model complexity to increase interpretability. The method learns these sparse local models by sampling (generating) new data points around existing ones and utilizes the complex model to predict the target property for these new points. SHAP uses additive feature attributions (linear models on binary variables) where the feature contributions are estimated using ideas from cooperative games in game theory.

Both LIME and SHAP require sampling of new data points. If care is not taken, the sampled data points are often unrealistic and might break underlying constraints in the dataset. Such constraints can incorporate, e.g., physical conservation laws such as energy conservation. Evaluating black box models on data dissimilar from the training data may lead to unreliable predictions [25] and nonsensical explanations [24].

SLISE [24, 26] is an XAI method explicitly designed to not rely on sampling new data, which automatically satisfies physical constraints. SLISE splits the data into a set of potential outliers and a set of non-outliers. This translates to explanations if we consider items with different local explanations as outliers (with respect to a selected data item). Non-outlier points then specify a linear model, while outlier points are ignored.

SLISEMAP builds on the idea of using actual data, as in SLISE [24], but adds the embedding as a manifold visualisation that acts as a global explanation for the underlying model. A common challenge when applying many manifold visualisation methods is that the resulting embedding depends on how we measure similarities between the data points. With SLISEMAP, we do not need to specify a distance metric in the high-dimensional data space. Rather, we measure how well the local model for one data item predicts the outcome of another data item.

## 3 Methods

In this section, we introduce the methods for this paper. We begin, in Sect. 3.1, with a brief overview of SLISEMAP [8]. and in Sect. 3.2, we describe how we use SLISEMAP and define several evaluation criteria in Sect. 3.3.

## 3.1 SLISEMAP

SLISEMAP [8] is a tool for visualising all local explanations in a dataset. Specifically, explanations are interpretable models that locally approximate the complex model, similar to [22–24]. SLISEMAP finds a low-dimensional embedding and local models (explanations) such that items with similar local models end up next to each other. In SLISEMAP, the local models are not limited to any single class; however, in this paper, we only consider simple linear regression and classification models. Formally, SLISEMAP solves the following problem:

**Problem 1** *Assume you are given a dataset of n data items $(x_1, y_1), \ldots, (x_n, y_n)$, where $x_i \in \mathbb{R}^m$ are the vectors of features and $y_i \in \mathbb{R}^o$ are the targets, and a radius $r \in \mathbb{R}_{>0}$. For every data item $i \in \{1, \ldots, n\}$, find the embedding $z_i \in \mathbb{R}^d$ and local model $f_i : \mathbb{R}^m \to \mathbb{R}^o$ that minimise the loss*

$$\mathcal{L} = \sum_{i=1}^{n}\sum_{j=1}^{n} \frac{e^{-\|z_i - z_j\|_2}}{\sum_{k=1}^{n} e^{-\|z_i - z_k\|_2}} l(f_i(x_j), y_j) \ + \ \sum_{i=1}^{n}\sum_{j=1}^{p} (\lambda_{lasso}|\mathbf{B}_{ij}| + \lambda_{ridge}\mathbf{B}_{ij}^2), \tag{1}$$

*where $\|\cdot\|_2$ is the Euclidean distance and $l(\cdot, \cdot)$ is a loss function for the local models under the constraint that*

$$\sum_{i=1}^{n}\sum_{k=1}^{d} z_{ik}^2 / n = r^2. \tag{2}$$

*The rows of $\mathbf{B} \in \mathbb{R}^{n \times p}$ contain the parameters for the local models $f_i$, where p is the number of parameters in the local interpretable models, and $\lambda_{lasso} \geq 0$ and $\lambda_{ridge} \geq 0$ are the parameters for Lasso and Ridge regularisation, respectively.*

Here we consider 2-dimensional embeddings ($d = 2$) with radius $r = 3.5$, as recommended by [8]. We use linear regression for $f_i$ and mean squared error as $l$ for regression problems. With classification problems, we use logistic regression for $f_i$ and Hellinger loss as $l$; see [8] for details. We optimise Eq (1) using LBFGS [27] combined with a greedy heuristic for escaping local optima in the SLISEMAP library [28, v1.5.2].

## 3.2 Workflow

We start by normalising the data into zero mean and unit variance. This step is optional but very common in machine learning, including manifold visualisation methods such as PCA and t-SNE. The normalisation improves the built-in lasso and ridge regularisation of the local models. Lasso regularisation also yields sparse solutions [29], making interpretation easier [7]. We apply SLISEMAP on the normalised data and use the $\chi$iplot visualisation platform for data exploration [30].

Manifold visualisation based on comparing items, including SLISEMAP [8] and t-SNE [12], tend to scale quadratically with the number of data items. A common solution [8] is to use subsampling for large datasets. However, SLISEMAP can utilise GPUs [8, 28], making plots with over 30,000 points, such as Fig 3, possible within a couple of minutes.

Some additional considerations are needed when applying XAI methods, including SLISEMAP. As mentioned in Sect. 2, for an explanation to be useful, the user must understand the components of the explanation [6]. Since we, in this paper, consider linear models, the features of $x$ must be interpretable. To this end, in Sect. 4, we represent the data in a tabular format with features that are interpretable to domain experts (other kinds of users might require different choices [6]).

Another consideration, mentioned in Sect. 2, is the number of features. In this paper, we only consider datasets with a small number of features (less than 40) and apply lasso regularisation to further reduce the amount of interpretation required by the user. Additionally, SLISEMAP

uses actual data instead of generating new data, so more features also require more data to fit the local models.

We also need the type of local model to suit the data. For the tabular datasets in this paper, linear models are understandable. But, for example, a linear model over individual colour channels in colour images is less meaningful. The black box model can also be chosen [31] or trained [32] to yield better explanations, but that is beyond the scope of this manuscript.

### 3.3 Performance measures

SLISEMAP produces an embedding and a local explanation for each data item. To draw any meaningful conclusions, we need to verify that the embedding and local models are reliable and not just random artefacts. Here we introduce three performance measures to test for various aspects of the solution: (i) permutation loss (a sanity check), (ii) local model stability (whether the set of local models is stable), and (iii) neighbourhood stability (whether points have a stable and unique embedding). We use these measures in Sect. 4.4.

**Permutation loss.** The *permutation loss* measures whether the SLISEMAP solution captures a connection between the features and the target variable. More specifically, denote by $\mathcal{L}_{permuted}$ the loss of Eq (1) on a SLISEMAP solution optimised after the target variable **y** has been randomly permuted, and by $\mathcal{L}$ the loss on the original unpermuted data. We define *permutation loss* as

$$\mathcal{M}_{permutation} = \mathcal{L}/\mathcal{L}_{permuted}. \tag{3}$$

Fig 2a shows the effect of the label permutation on SLISEMAP. Naturally, the SLISEMAP model (orange lines) trained on permuted data (orange circles) is worse at predicting the unknown function (blue crosses). In general, we expect $\mathcal{M}_{permutation} < 1$; otherwise, the SLISEMAP has not been able to capture any relations between the features and the target variable. The permutation loss acts as a sanity check for the solution.

**Local model stability.** If we train SLISEMAP solutions with subsets of different items, they should provide similar sets of local models. Otherwise, the local models do not provide a sufficient approximation of the data-generating process under study, e.g., due to insufficient data,

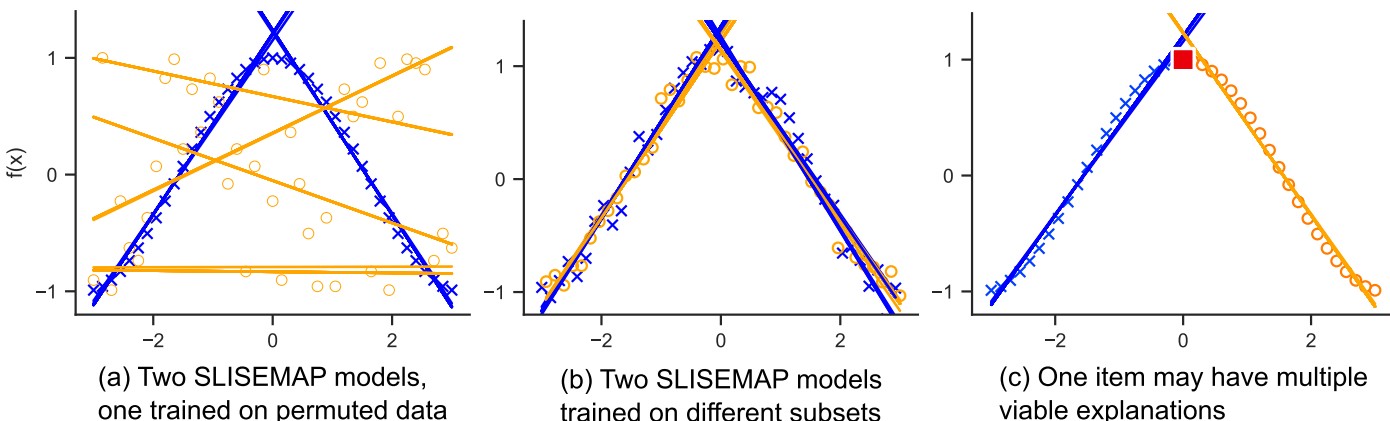

(a) Two SLISEMAP models, one trained on permuted data

(b) Two SLISEMAP models trained on different subsets

(c) One item may have multiple viable explanations

**Fig 2. Demonstrating the performance measures described in Sect. 3.3.** In the left figure, we show two SLISEMAP models, one trained on an unknown function (blue crosses) and the other trained on permuted labels (orange circles). In the middle figure, two SLISEMAPs are trained on non-overlapping noisy samples from an unknown function. The local models (blue and orange lines) are very similar, and the local model stability between the two models is high. In the right figure, we demonstrate the phenomenon of alternative explanations: a point at $x = 0$ (red square) could be explained almost equally well by either local model. (a) Two SLISEMAP models, one trained on permuted data. (b) Two SLISEMAP models trained on different subsets. (c) One item may have multiple viable explanations.

overfitting, or an unsuitable type of local model (for the data). Hence, we need a high *local model stability* to make further deductions.

Assume we have obtained two datasets of the same size, e.g., by resampling $n$ data points without replacement from the original data, resulting in two sets of local models $\{f_1, \ldots, f_n\}$ and $\{f'_1, \ldots, f'_n\}$. To compare the similarity of the local model populations, we use the Hungarian algorithm to find a permutation $\pi$ such that the sum of distances is minimised:

$$\mathcal{M}_{models} = 1 - \min_{\pi} \sum_{i=1}^{n} D(f_i, f'_{\pi(i)}) \Big/ Z, \tag{4}$$

where $D(f_i, f'_{\pi(i)}) = \|\mathbf{B}_{i,\cdot} - \mathbf{B}_{\pi(i),\cdot}\|_2$ is the Euclidean distance between the coefficients of the local models and $Z = \sum_{i=1}^{n} \sum_{j=1}^{n} D(f_i, f'_j)/n$.

Fig 2b shows a simple example of a setting with high local model stability. Two SLISEMAP models are trained with non-overlapping samples (depicted as blue crosses and orange circles) from an unknown function: in this case, a noisy version of the function shown in Fig 1. Fig 2b shows how the SLISEMAP models find very similar sets of local explanations, and accordingly, the local model stability is high, $\mathcal{M}_{models} \approx 0.87$.

**Neighbourhood stability.** The *neighbourhood stability* measures the stability of the neighbourhoods with respect to the resampling of the data, i.e., is the relative location of individual data points stable under resampling? To compute neighbourhood stability, we sample two datasets of size $n$, which share 50% of the data items. Denote by $\mathcal{S}$ the set of points shared by these two datasets, and by $\{z_1, \ldots, z_n\}$ and $\{z'_1, \ldots, z'_n\}$ the points in the two embeddings. We define the neighbourhoods of point $i$ in the first embedding by $N(i) = \{j \in \mathcal{S} \mid \|z_i - z_j\|_2 \leq 1\}$ and $N'(i) = \{j \in \mathcal{S} \mid \|z'_i - z'_j\|_2 \leq 1\}$ in the second embedding. The *neighbourhood stability* is defined as an average Jaccard similarity between the neighbourhoods:

$$\mathcal{M}_{neighbourhood} = \frac{1}{|\mathcal{S}|} \sum_{i \in \mathcal{S}} |N(i) \cap N'(i)| \ / \ |N(i) \cup N'(i)| \tag{5}$$

If an embedding reflects some real structure in the data, we would assume that the embedding would be stable with respect to resampling. A number of effects can decrease neighbourhood stability. First, sampling a new, different subset of data might reveal structures not present in the previous subset. This would indicate that more data is required to find a stable embedding. Second, there might not be a meaningful structure that SLISEMAP can find. Third, for many items, there may exist a number of alternative explanations [8] (local models) that predict the label associated with a given item nearly as well. Changes in the training dataset can easily cause the explanation for these items to switch to an alternative with a different embedding. Consider, e.g., the point near $x = 0$ in Fig 2c. Both local models explain this item nearly equally well.

**Explanation quality.** In addition to the metrics described above, we want to measure the quality of the explanations produced by SLISEMAP. To measure explanation quality, we use three metrics found in the literature: local loss, local loss on nearest neighbours and coverage. These metrics are described in [8], and we have included their definitions in the Sect. S3 in S1 File. Some metrics require choosing additional parameters; in such cases, we use values found in [24].

Local loss measures how well the local models fit their targets (or, in the case of nearest neighbours' local loss, those of their neighbours). Coverage counts how many other all other data items the local models generalise to. We also count how many nearest neighbour the local models cover.

For comparison, we calculate these values for other widely used manifold visualisation (PCA, t-SNE and UMAP) and XAI (SHAP, LIME and SLISE) methods. To produce explanations for the manifold visualisation methods, we first calculate an embedding and then train local models based on that embedding (similar to SLISEMAP). For the manifold visualisation methods (PCA, t-SNE, UMAP and SLISEMAP), the nearest neighbours (in the measures) are chosen based on the embedding. The other XAI methods (LIME, SHAP and SLISE) do not produce an embedding and thus the nearest neighbours are chosen in the original data space.

**Visual inspection.** SLISEMAP is foremost designed as a tool for investigating and understanding black box models. Therefore, the main way to analyse the solutions is through plots [30]. We demonstrate how to analyse SLISEMAP solutions for multiple datasets in Sect. 4 and how domain expertise can help interpret and validate the solutions.

Analysing potentially thousands of local models is not feasible. Hence, we often cluster the local models using K-MEANS clustering on the coefficients of the local models, i.e., the rows of matrix **B**. If a suitable number of clusters is chosen, the cluster centroids are a good proxy for most local models. However, data items in lower-density areas of the embedding might have very different local models from the cluster centroids. Since they also have small neighbourhoods, the local models are at risk of overfitting, and care should be taken when interpreting these points. Notice that we use K-MEANS clustering after SLISEMAP only to help visualise and interpret the results.

## 4 Use cases

This section demonstrates how SLISEMAP can be applied to physical datasets and how we analyse and verify the solutions using domain knowledge and the metrics from above. The datasets and source code used in the paper are available at https://github.com/edahelsinki/paper-slisemap-physical. In Sect. 4.1, we study a dataset from atmospheric science; in Sect. 4.2, a dataset from high energy physics, and in Sect. 4.3, a dataset about organic chemistry. Finally, in 4.4, we evaluate the solutions using the performance measures from Sect. 3.3.

### 4.1 Atmospheric relevant organic molecules: GeckoQ

We applied SLISEMAP to 31,637 atmospheric relevant organic compounds contained in the *GeckoQ* dataset [33], collected by one of the authors. Each molecule has a variety of properties, such as the number of specific functional groups, *saturation vapour pressure* ($p_{Sat}$) and *topological fingerprint* (TopFP) descriptor [34, 35].

The $p_{Sat}$ is a measure of the affinity of a molecule to condense into the liquid phase or to remain in the gas phase. This property is relevant in atmospheric science because low-volatile organic compounds are known to be driving factors for particle formation in the atmosphere [36]. The GeckoQ $p_{Sat}$ have been calculated with the *Conductor-like Screening Model for real solvents* (COSMO-RS) [37, 38], a quantum chemistry-based method, and will be the (log-transformed) target variables in following SLISEMAP embeddings. A subset of molecular properties was chosen as *interpretable features* to construct the SLISEMAP embeddings. A list of all explainable features can be found in the Sect. S1 in S1 File.

Fig 3 shows a SLISEMAP embedding of the GeckoQ data. The local model coefficients generally match chemical expectations, meaning *functional groups* (FGs) that are especially known to lead to a low $p_{Sat}$ have the most negative coefficients in their local models: hydroxyl, hydroperoxide, carboxylic acid, and carbonylperoxyacids. These FGs can form hydrogen bonds, the strongest type of inter-molecular dipole-dipole interactions, and thus, the molecules are particularly strongly bound within the liquid phase, which leads to a low $p_{Sat}$. The incidence of each FG by cluster can be found in the S1 Fig in S1 File.

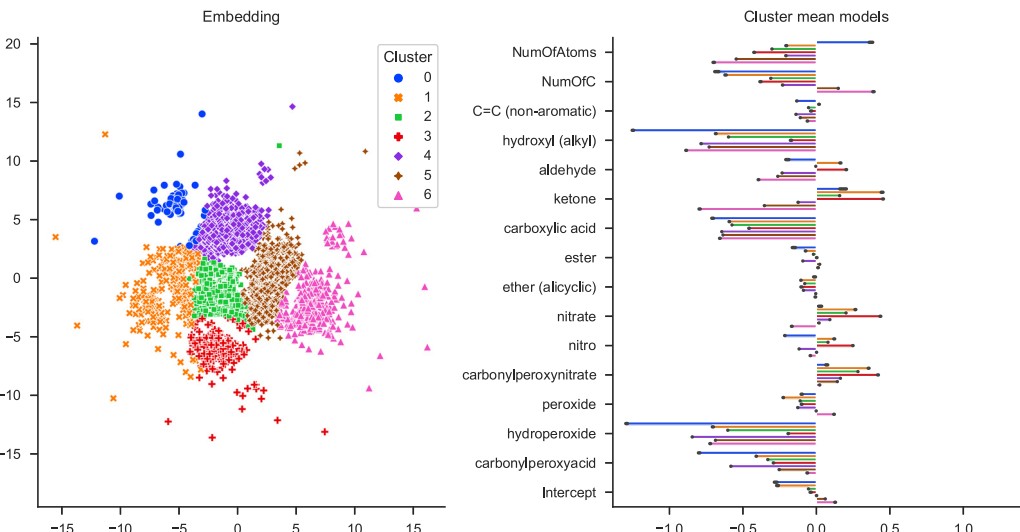

**Fig 3. The SLISEMAP embedding of the GeckoQ data in the left panel.** The number of clusters (seven) was chosen via visual inspection. The right panel includes the average local coefficients of each cluster.

To further substantiate the SLISEMAP embedding's agreement with the chemical expectancies, we chose cluster 1 (orange) and cluster 6 (pink), two visually distinct clusters for detailed analysis. In cluster 1, the median target (non-normalised) is $2.15 \cdot 10^{-9}$ mbar (1686 molecules), whereas cluster 6 contains molecules with a higher median $p_{Sat}$, $5.75 \cdot 10^{-6}$ mbar (1862 molecules). The median $p_{Sat}$ for all the data is $1.55 \cdot 10^{-6}$ mbar. Fig 4a depicts the fraction of molecules in the chosen clusters and the overall data containing FGs forming hydrogen bonds. Cluster 1 contains considerably more carboxylic acid groups than cluster 6 and the overall data, which can be directly linked to its particularly low median $p_{Sat}$. Additionally, cluster 1 shows a higher incidence of hydroxyl groups, while cluster 6 has a higher fraction of carbonyl-peroxyacid groups. Both clusters have a similar fraction of hydroperoxide groups.

Differences between clusters also become apparent through binning the data points in the SLISEMAP embedding and analysis of the median (normalised) target values in the bins (cf. Fig 4b). Generally, regions of low and high $p_{Sat}$ emerge. These regions roughly correspond to the clusters determined by SLISEMAP in Fig 3, which is the most distinct in the points corresponding to clusters 1 and 6. For comparison, Fig 4c depicts the binned logarithm of the median target for a t-SNE embedding, an alternative way of visualising data projected into a two-dimensional space that is popular in the field of chemoinformatics. Unlike the SLISEMAP embedding, the t-SNE does not display distinct regions of low and high $p_{Sat}$.

Furthermore, the explanation quality metrics (see Table 1) show that SLISEMAP produces truly local models while the t-SNE embedding produces more general ones. This reinforces how SLISEMAP is able to prioritise patterns related to the target when building the embedding, whereas t-SNE, due to its unsupervised nature, cannot group molecules with similar behaviour together.

## 4.2 Elementary particle jets

This dataset contains simulated LHC proton-proton collisions [39]. The collisions create elementary particles such as *quark*s and *gluon*s. Quarks and gluons decay into cascades of stable

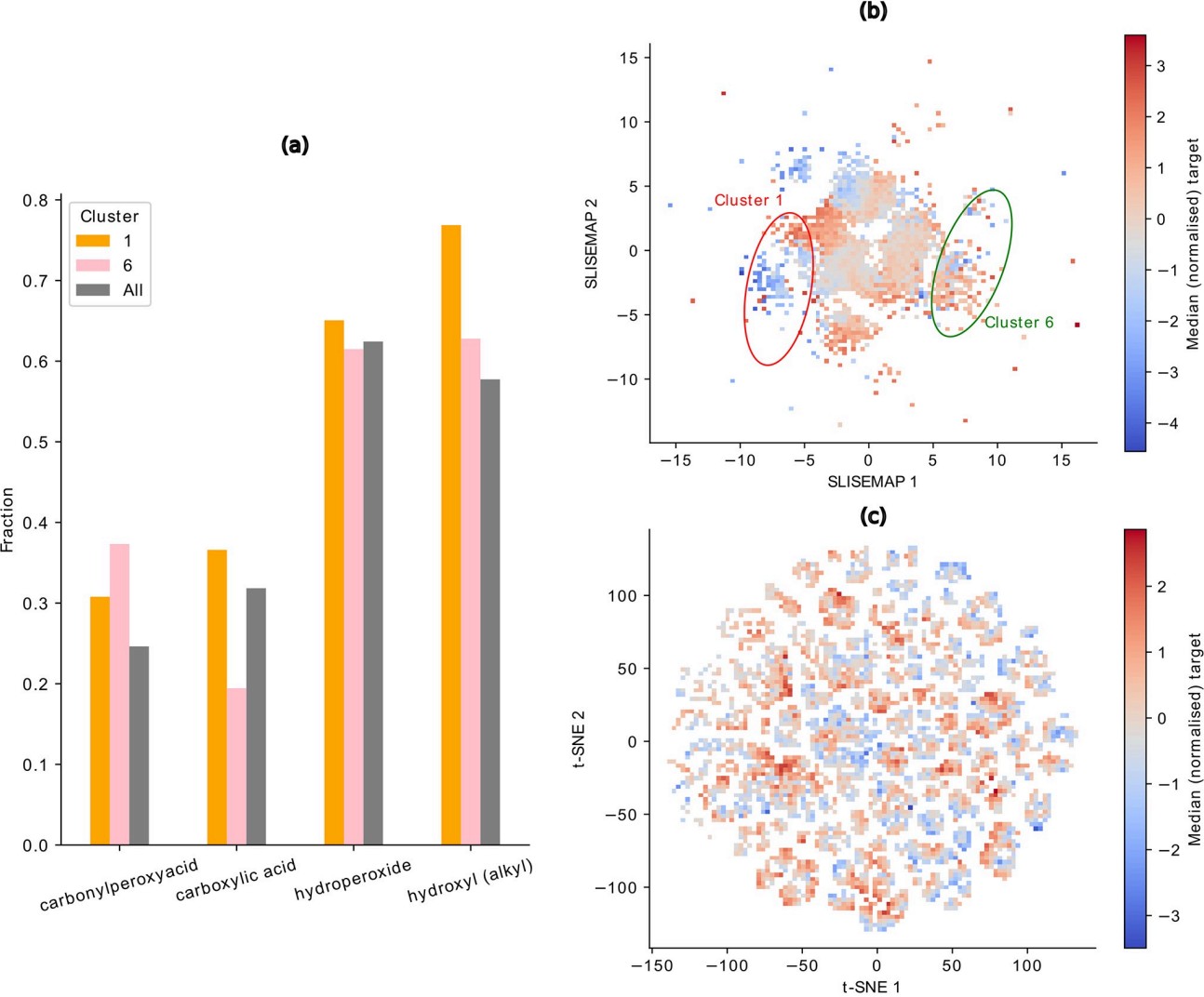

**Fig 4.** (a) Fraction of molecules that contain at least one FG of hydroxyl, hydroperoxide, carboxylic acid, carbonylperoxyacid grouped by clusters 1, 6 and all the clusters. (b) SLISEMAP and (c) t-SNE embedding, where the data points are binned, and the colour map corresponds to the median normalised target of the bins. Clusters 1 and 6 are encircled.

particles, called jets, before being detected. The classification task is to determine if the particle that created a jet was a quark or a gluon [40].

To get probabilities for the jets, we train a random forest classifier [41] with 100 trees and 50 leaves per tree. When applying SLISEMAP [8] on this dataset, we use logistic regression as the local models $f_i$ in Eq (1).

The resulting solution can be seen in Fig 5. The local models match the underlying quantum chromodynamics [40]. Wider jets (high `jetGirth` and `QG_axis2`) and jets with more particles (`QG_mult`) are more gluon-like. Splitting the multiplicity based on charge (`jetChargedMult` and `jetNeutralMult`) should not offer any more information. The total momentum (`jetPt`) usually does not matter, except for higher energies where quarks are more likely. How much of the total momentum parallels the jet (`QG_ptD`) is also helpful in finding quarks.

**Table 1. Comparison of explanation measures for various manifold visualisation and XAI methods for the datasets described in this paper; computed as described in [8].** Bold values indicate the best performance and the error bounds the standard deviation with respect to the resampling of the data. The time column measures the total training time as well as the time to provide explanations for 5000 items without a GPU.

| Method | Time (s) ↓ | Local loss ↓ | NN Local loss ↓ | Coverage ↑ | NN Coverage ↑ |
|---|---|---|---|---|---|
| **Dataset: GeckoQ** | | | | | |
| SLISEMAP | 4461.663 ± 1822.30 | 0.020 ± 0.01 | **0.020 ± 0.01** | 0.235 ± 0.01 | **0.912 ± 0.04** |
| PCA | **42.424 ± 2.46** | 0.245 ± 0.03 | 0.286 ± 0.01 | 0.293 ± 0.00 | 0.315 ± 0.01 |
| t-SNE | 48.991 ± 3.34 | 0.242 ± 0.03 | 0.287 ± 0.01 | 0.292 ± 0.00 | 0.313 ± 0.01 |
| UMAP | 176.567 ± 4.67 | 0.248 ± 0.03 | 0.292 ± 0.01 | 0.293 ± 0.00 | 0.310 ± 0.01 |
| SLISE | 10353.657 ± 130.69 | **0.000 ± 0.00** | 0.391 ± 0.02 | **0.315 ± 0.00** | 0.288 ± 0.01 |
| **Dataset: Jets** | | | | | |
| SLISEMAP | 12267.437 ± 3558.67 | **0.000 ± 0.00** | **0.000 ± 0.00** | 0.694 ± 0.01 | **0.999 ± 0.00** |
| PCA | 204.405 ± 9.31 | 0.001 ± 0.00 | 0.001 ± 0.00 | 0.715 ± 0.01 | 0.976 ± 0.00 |
| t-SNE | **153.282 ± 6.05** | 0.001 ± 0.00 | 0.001 ± 0.00 | 0.700 ± 0.01 | 0.969 ± 0.00 |
| UMAP | 216.345 ± 6.62 | 0.001 ± 0.00 | 0.001 ± 0.00 | **0.770 ± 0.01** | 0.928 ± 0.01 |
| LIME | 313.953 ± 2.36 | 0.004 ± 0.00 | 0.012 ± 0.00 | 0.277 ± 0.01 | 0.397 ± 0.02 |
| SHAP | 2395.418 ± 37.31 | **0.000 ± 0.00** | 0.008 ± 0.00 | 0.327 ± 0.01 | 0.592 ± 0.02 |
| SLISE | 5369.622 ± 219.57 | **0.000 ± 0.00** | 0.003 ± 0.00 | 0.690 ± 0.01 | 0.831 ± 0.02 |
| **Dataset: QM9** | | | | | |
| SLISEMAP | 4890.629 ± 1294.40 | 0.025 ± 0.00 | **0.035 ± 0.02** | 0.253 ± 0.01 | **0.821 ± 0.03** |
| PCA | **188.364 ± 12.83** | 0.203 ± 0.07 | 0.226 ± 0.01 | 0.293 ± 0.00 | 0.352 ± 0.01 |
| t-SNE | **190.734 ± 24.32** | 0.196 ± 0.07 | 0.212 ± 0.01 | 0.285 ± 0.00 | 0.362 ± 0.01 |
| UMAP | 322.026 ± 20.69 | 0.219 ± 0.08 | 0.214 ± 0.01 | 0.288 ± 0.00 | 0.367 ± 0.01 |
| LIME | 330.723 ± 5.96 | 1.369 ± 0.42 | 1.620 ± 0.40 | 0.110 ± 0.02 | 0.108 ± 0.02 |
| SHAP | 7135.664 ± 1420.74 | **0.000 ± 0.00** | 0.724 ± 0.05 | 0.185 ± 0.00 | 0.223 ± 0.01 |
| SLISE | 11119.356 ± 157.68 | **0.000 ± 0.00** | 0.361 ± 0.03 | **0.381 ± 0.00** | 0.394 ± 0.01 |

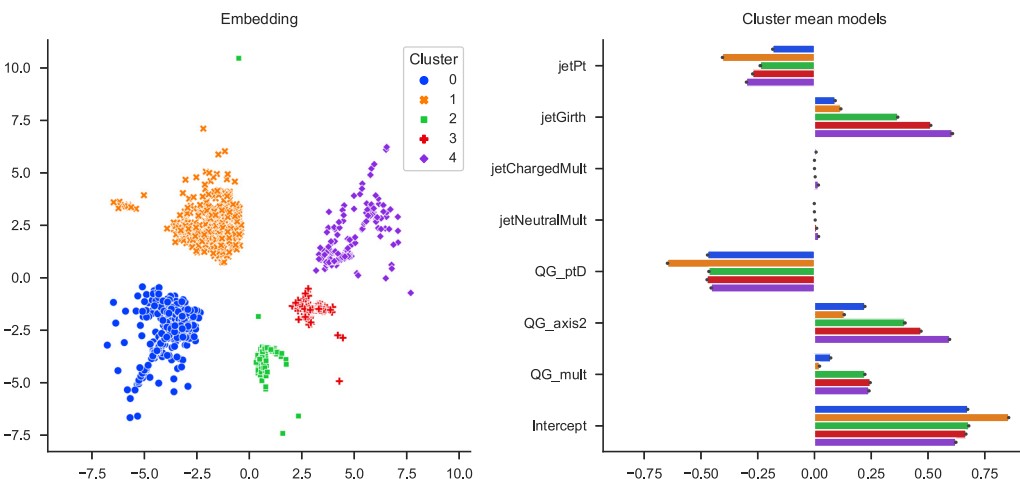

**Fig 5. SLISEMAP solution for 10 000 jets from the particle jets dataset.** The data items have been clustered according to the local models. The coefficients for all local models follow physical theory but with varying magnitudes.

Even though all local models match the theory, there are some differences in magnitudes. E.g., the blue cluster 0, in Fig 5, contains most of the highest energy jets. Since multiplicity and, indirectly, the spread is also dependent on the energy, these variables are less helpful in this subset. In contrast, the orange cluster 1 contains many jets easily classified as gluons based on `QG_ptD` alone.

Summarising, the SLISEMAP explanations are consistent with the physical knowledge (gluon jets are generally wider) and provide additional insights into the features used by the random forest classifier.

## 4.3 Small organic molecules: QM9

Finally, we applied SLISEMAP to the molecules in the QM9-dataset [42, 43], a data set of 133,814 small organic molecules. We utilised HOMO energies obtained from [44, 45] as targets and created interpretable features with the Mordred molecular descriptor calculator [46]. Here, we will focus on qualitative analysis based on the SLISEMAP embedding and omit a more technical chemical analysis. Fig 6 displays a similar embedding structure to the GeckoQ embedding: the bulk of the data points group around a few centres, and a small number of data points are spread out. However, unlike GeckoQ, the clusters are much more distinct. Setting the number of clusters to five, we can see a large central cluster flanked by four others. For the central cluster, the number of hydrogen bond donors (`nHBDon`) has relatively lower importance than for the other clusters, as can be seen in the right panel in Fig 6. The orange cluster is distinguished from the others by the hybridisation ratio (`HybRatio`), which, conversely, is regularised to zero for the green cluster. The number of oxygen bonds (`nBondsO`) is an important feature of the small yet distinct purple cluster.

## 4.4 Evaluation of the solutions

It is essential to evaluate whether the resulting visualisations contain useful information or whether we are just visualising random noise. The three performance measures described in Sect. 3.3 are designed to evaluate various aspects of the visualisations. We depict the performance measures as a function of the sample size in Fig 7. For each measure and subsampled size, we train ten models and average over them to get the final performance measure value.

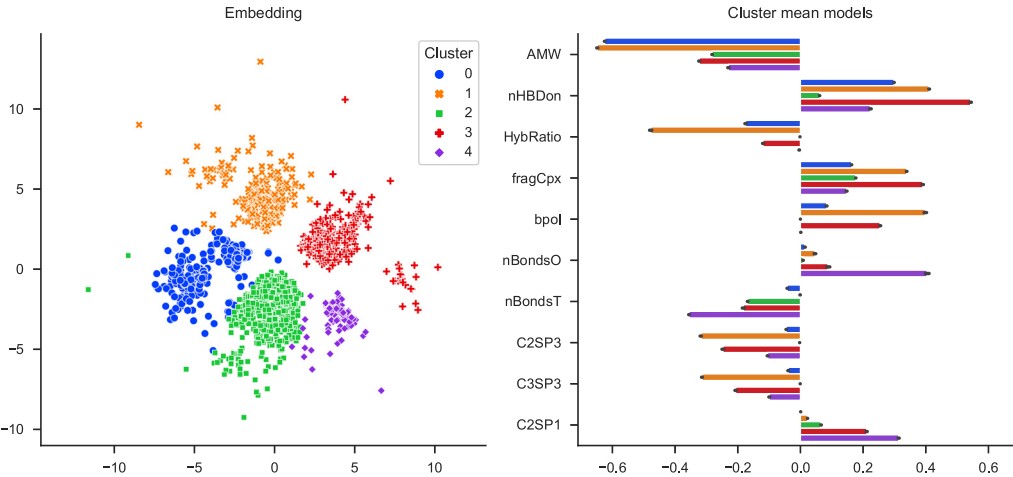

**Fig 6. SLISEMAP embedding for the QM9 data set (10,000 molecules), clustered with 5 clusters.** The right panel shows the ten most influential features for predicting the target (HOMO energy).

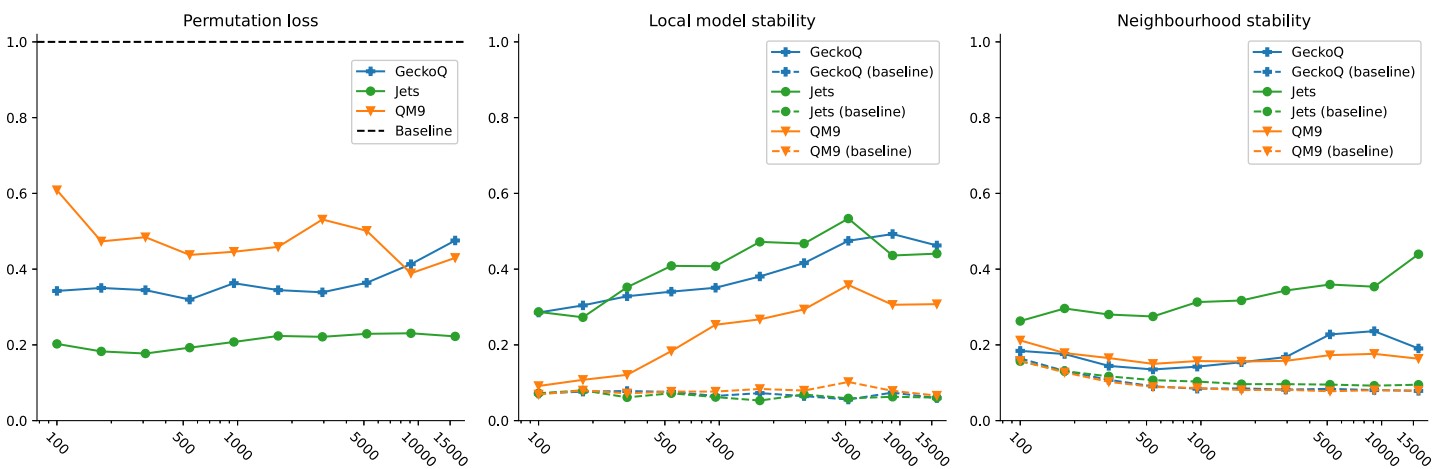

**Fig 7. Permutation loss, local model stability and neighbourhood stability (Sect. 3.3) for the use cases as a function of (resampled) dataset size.** For permutation loss, smaller values are better; for local model and neighbourhood stabilities, higher values are better. Dashed lines indicate reference values for baseline datasets where the target variables have been permuted randomly.

To provide a baseline for the stability measures, we also calculate the measure between SLISE-MAPS trained on actual data and those trained on data with randomly permuted target variables.

The permutation loss for each dataset remains less than one (the permuted baseline), assuring that SLISEMAP has learned from the data. The local model stability shows similar behaviour, rapidly exceeding the baseline as sample size increases, especially for the particle jets dataset. The neighbourhood stability is reasonable for the particle jets data set but worse for the molecular datasets.

Comparison of explanation quality in Table 1 shows that SLISEMAP produces comparable or better explanations to other XAI or manifold visualisation methods. The table is generated based on taking 5000 samples from each of the datasets.

As the manifold visualisation methods are unsupervised, their ability to generate accurate explanations is poorer than SLISEMAP with all of the three datasets. Conversely, the XAI methods SLISE and SHAP fit an exact local model (zero local loss) to each of the data items. However, SLISEMAP attains very low local losses, and the explanations can also predict nearby points in the SLISEMAP embedding well, as demonstrated by the low local loss and high coverage for the nearest neighbours. As SLISEMAP explanations are more specialised, their global coverage is slightly less than the manifold visualisations. Nonetheless, SLISEMAP still attains a higher global coverage than LIME and SHAP, meaning that explanations from SLISEMAP both explain the individual items and generalise better. Unlike the other XAI methods, SLISEMAP also produces a local explanation for all of the data items simultaneously, whereas the others can only produce one explanation at a time.

## 4.5 Discussion

Our aim with SLISEMAP is two-fold: we want to simultaneously provide local explanations and an informative embedding. Table 1 shows that SLISEMAP has unique benefits compared to other, widely used manifold visualisation and XAI methods. Furthermore, the embeddings in each case are non-random (permutation loss is less than one), and the sets of local models (as measured by local model stability) are informative.

For the particle jets dataset, we can show that the SLISEMAP model provides valuable information both qualitatively in the form of sensible embedding and quantitatively with our proposed explanation quality metrics. While the neighbourhood stability for the molecular datasets is lower in comparison, we show how SLISEMAP provides a more meaningful embedding related to the prediction task than t-SNE with the GeckoQ data.

The lower neighbourhood stability of the molecular datasets shows that a single molecule can have several alternative local explanations, as discussed in Sect. 3.3 and [8]. We argue that these alternative explanations are inherent to all local explanation methods, including SLISEMAP. For example, data items near the intersection of the local model hyperplanes could be explained almost equally well by both intersecting models, as illustrated in Fig 2c. As a practical consequence, one should be careful when, e.g., studying the embedding of individual molecules (as a molecule could potentially be allocated to multiple clusters). However, even for the molecular datasets, the overall sets of local models are stable.

The decrease in neighbourhood stability might be compounded by the molecular datasets themselves and their associated black box models. Indeed, research [31] suggests that the performance of local linear explanation models such as SLISEMAP partially depends on the type of black box model. It should be noted that the explanation quality metrics in Table 1 are high for all methods with the particle jets dataset. This suggests that the particle jets data is easier to explain.

## 5 Conclusions

In this paper, we apply SLISEMAP [8], a recent XAI and visualisation method, on three scientific datasets. Using domain knowledge, we verify that the explanations are consistent with the chemical and physical theories.

SLISEMAP captures the chemical expectations set for the GeckoQ data. The local models match the different natures of the functional groups. Further, SLISEMAP can find underlying structures in the atmospheric data and reflect the relationship between the target and the features, which are generally known but hardly quantifiable. The local models in the particle jets dataset also adhere to (known) physics. However, the subgroups with slightly different models were new but, after analysis, justifiable. For the QM9 dataset, structure emerges in the SLISEMAP embedding with distinctively behaving clusters of molecules with different local explanations for each of the clusters.

In Sect. 3.3, we introduce stability measures for the SLISEMAP embedding. In Sect. 4.4, we show that the collection of explanations (local models) found by the SLISEMAP are informative and stable for all our use cases. Moreover, Table 1 shows that SLISEMAP captures patterns for predicting the target value better than other manifold visualisation and XAI methods widely used in these domains. However, our stability measures also reveal that the explanations for individual data items are often not unique, a fact that SLISEMAP (unlike most other local explanation methods) makes apparent [8].

In the future, it would be interesting to analyse the alternative local explanations in depth. Furthermore, our results suggest that while individual points may have several explaining local models, the overall distribution of local models is stable over random resampling of the data. Constructing these sets of plausible local models and, by extension, approximating the overall distribution could allow for uncertainty quantification. Finally, the fact that we are able to approximate the black box models with multiple local models suggests that we might be able to replace the black box models with a more interpretable procedure.

In conclusion, we have demonstrated a workflow for applying SLISEMAP and shown how SLISEMAP can provide physically sound insight for analysing complex physical datasets.

## Supporting information

**S1 File.**
(PDF)

## Acknowledgments

We thank Dr. Jarmo Mäkelä, Prof. Patrick Rinke, and Dr. Hilda Sandström for helpful discussions.

## Author Contributions

**Conceptualization:** Lauri Seppäläinen, Anton Björklund, Vitus Besel, Kai Puolamäki.

**Data curation:** Anton Björklund, Vitus Besel.

**Formal analysis:** Lauri Seppäläinen, Anton Björklund, Vitus Besel, Kai Puolamäki.

**Funding acquisition:** Kai Puolamäki.

**Investigation:** Lauri Seppäläinen, Anton Björklund, Vitus Besel.

**Methodology:** Lauri Seppäläinen, Anton Björklund, Vitus Besel, Kai Puolamäki.

**Project administration:** Kai Puolamäki.

**Software:** Lauri Seppäläinen, Anton Björklund.

**Supervision:** Kai Puolamäki.

**Validation:** Lauri Seppäläinen, Anton Björklund.

**Visualization:** Lauri Seppäläinen, Anton Björklund, Vitus Besel.

**Writing – original draft:** Lauri Seppäläinen, Anton Björklund, Vitus Besel, Kai Puolamäki.

**Writing – review & editing:** Lauri Seppäläinen, Anton Björklund, Vitus Besel, Kai Puolamäki.

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
