## [Decision Letter · Decision Letter 0]

26 Nov 2023

PONE-D-23-35222

Using Slisemap to interpret physical data

PLOS ONE

Dear Dr. Seppäläinen,

Thank you for submitting your manuscript to PLOS ONE. After careful consideration, we feel that it has merit but does not fully meet PLOS ONE’s publication criteria as it currently stands. Therefore, we invite you to submit a revised version of the manuscript that addresses the points raised during the review process.

After a preliminary review, I find your work intriguing and valuable to our field. However, to move forward in the review process, I would appreciate your detailed responses to the following queries as well as the reviewer's queries:

Detailed Performance Evaluation: The performance measures (permutation loss, local model stability, and neighbourhood stability) you used are intriguing. Can you offer more comprehensive insights into how these measures evaluate the effectiveness of SLISEMAP in visualizing and interpreting complex datasets?

Varied Field Applications: Your paper demonstrates the application of SLISEMAP across atmospheric science, high energy physics, and organic chemistry. Could you provide further details or additional examples illustrating how SLISEMAP's application and effectiveness vary across these fields?

Comparison with Existing Methods: Given the rise of supervised learning in physical sciences, how does SLISEMAP complement or enhance existing methods for interpreting complex models in these disciplines?

Considering the diverse range of applications of SLISEMAP, can you discuss its scalability with respect to dataset size and complexity? Are there particular limitations or requirements for data that SLISEMAP is best suited for?

We look forward to receiving your revised manuscript.

Kind regards,

Jayesh Soni

Academic Editor

PLOS ONE

Journal Requirements:

   "We thank the Research Council of Finland (decisions 346376 (LS, KP), 345704 (KP), 337549 (VB) and 34636 (VB)), and the Doctoral Programme in Computer Science at University of Helsinki (AB) for funding, and the Finnish Computing Competence Infrastructure (FCCI) for supporting this project with computational resources.

Open access funded by Helsinki University Library." 

Additional Editor Comments:

After a preliminary review, I find your work intriguing and valuable to our field. However, to move forward in the review process, I would appreciate your detailed responses to the following queries:

1. Methodological Distinction: Your paper introduces the application of SLISEMAP in manifold visualization combined with explainable artificial intelligence (XAI). Could you provide a more detailed comparison of SLISEMAP with other manifold visualization methods like PCA or t-SNE, particularly in terms of handling high-dimensional data in physical sciences?

2. Impact of Data Normalization: The normalization of data to zero mean and unit variance before applying SLISEMAP is an important step. Could you elaborate on how this impacts both the interpretability and accuracy of your method, considering the varied nature of datasets in physical sciences?

3. Detailed Performance Evaluation: The performance measures (permutation loss, local model stability, and neighbourhood stability) you used are intriguing. Can you offer more comprehensive insights into how these measures evaluate the effectiveness of SLISEMAP in visualizing and interpreting complex datasets?

4. Varied Field Applications: Your paper demonstrates the application of SLISEMAP across atmospheric science, high energy physics, and organic chemistry. Could you provide further details or additional examples illustrating how SLISEMAP's application and effectiveness vary across these fields?

5. Comparison with Existing Methods: Given the rise of supervised learning in physical sciences, how does SLISEMAP complement or enhance existing methods for interpreting complex models in these disciplines?

6. Handling Similarities Between Data Points: The introduction highlights a challenge in manifold visualization regarding the dependency of embedding on the measurement of similarities between data points. How does SLISEMAP address this issue, and what specific methods are employed to measure and interpret these similarities?

Reviewers' comments:

Reviewer's Responses to Questions

**Comments to the Author**

1. Is the manuscript technically sound, and do the data support the conclusions?

Reviewer #1: Yes

Reviewer #2: Yes

2. Has the statistical analysis been performed appropriately and rigorously? 

Reviewer #1: Yes

Reviewer #2: Yes

3. Have the authors made all data underlying the findings in their manuscript fully available?

Reviewer #1: Yes

Reviewer #2: Yes

4. Is the manuscript presented in an intelligible fashion and written in standard English?

Reviewer #1: Yes

Reviewer #2: Yes

5. Review Comments to the Author

Reviewer #1: The paper mentions the use of SLISEMAP for manifold visualization combined with explainable artificial intelligence (XAI). Could you elaborate on how SLISEMAP differs from other manifold visualization methods, such as principal component analysis (PCA) or t-SNE, specifically in the context of handling high-dimensional data in physical sciences?

You mention normalizing the data to zero mean and unit variance before applying SLISEMAP. How does this normalization impact the interpretability and accuracy of the SLISEMAP method, especially considering the diverse nature of datasets in physical sciences?

Reviewer #2: Comparison with Traditional Methods: Given the increasing application of supervised learning in physical sciences, how does SLISEMAP improve or complement existing methods for interpreting complex models in these fields?

Handling of Data Features: The introduction section mentions that a challenge in manifold visualization is how the embedding depends on measuring similarities between data points. How does SLISEMAP address this challenge, and what methods does it use to measure and interpret these similarities?

6. PLOS authors have the option to publish the peer review history of their article (what does this mean?). If published, this will include your full peer review and any attached files.

Reviewer #1: No

Reviewer #2: No

---

## [Author Response · Author response to Decision Letter 0]

8 Jan 2024

Dear Editor and Reviewers,

Thank you for the insightful reviews. In addition to your suggestions, we have decided to re-write the introduction (Section 1) to make it more clear and reduce the overlap with Section 2. We also introduce a new Figure 1 to give a simple, intuitive example of how SLISEMAP works, which we also use when describing the motivation behind the performance measures in Section 3.3. A new Section 3.2 is introduced, where we discuss important considerations when using SLISEMAP, such as data format, normalisation and complexity. Additionally, we have updated the QM9 dataset analysis in Section 4.3 with a newer set of interpretable features (we have retrained and re-evaluated the results).

Sincerely,

Lauri Seppäläinen, Anton Björklund, Vitus Besel and Kai Puolamäki

Editor's comments:

1. Detailed Performance Evaluation: The performance measures (permutation loss, local model stability, and neighbourhood stability) you used are intriguing. Can you offer more comprehensive insights into how these measures evaluate the effectiveness of SLISEMAP in visualizing and interpreting complex datasets?

Answer: We have extended Section 3.3 with discussions about the implications of good/bad performance and added Figure 2, demonstrating the intuition of the measures on simple examples (similar to the new Figure 1). We also discuss the results of the measures in more detail in Section 4.4 and the implications for applicability in Section 3.

2. Varied Field Applications: Your paper demonstrates the application of SLISEMAP across atmospheric science, high energy physics, and organic chemistry. Could you provide further details or additional examples illustrating how SLISEMAP's application and effectiveness vary across these fields?

Answer: We have extended the comparison and analysis in Section 4.4 and added Section 4.5, further discussing the implications for different applications. We added more explicit discussion on the qualitative effectiveness of SLISEMAP as a tool in physical sciences, supported by the quantitative results from various explanation quality and stability metrics. We have also added a discussion about important considerations when applying SLISEMAP to Section 3.2.

3. Comparison with Existing Methods: Given the rise of supervised learning in physical sciences, how does SLISEMAP complement or enhance existing methods for interpreting complex models in these disciplines?

Answer: Many local explanation methods rely on (randomly) generating new data. This can yield unrealistic, out-of-distribution data, which can lead to unreliable predictions and explanations [2]. With SLISEMAP, we avoid this issue by not generating new data. Furthermore, the collection of local explanations forms a sort of global explanation (through the embedding) and also lets us find alternative local explanations (for one data item). We have slightly clarified this topic in Section 2 and added discussions in Sections 3.2 and 3.3. We also extended Table 1 with a comparison to a couple of related local explanation methods and added the corresponding discussion to Section 4.4.

4. Considering the diverse range of applications of SLISEMAP, can you discuss its scalability with respect to dataset size and complexity? Are there particular limitations or requirements for data that SLISEMAP is best suited for?

Answer: We have extended Section 3.2 with important considerations for the data when using XAI in general, and SLISEMAP in particular. We also added a discussion about the quadratic complexity to Section 3.2 with proposed procedures for large datasets: subsampling and GPU acceleration, see

[1] for a detailed evaluation. Furthermore, Table 1 now contains a time column.

Detailed reviewer comments:

1. Methodological Distinction: Your paper introduces the application of SLISEMAP in manifold visualization combined with explainable artificial intelligence (XAI). Could you provide a more detailed comparison of SLISEMAP with other manifold visualization methods like PCA or t-SNE, particularly in terms of handling high-dimensional data in physical sciences?

Answer: As answered above, in Section 3.2, we now discuss the quadratic scaling and include times in Table 1.

2. Impact of Data Normalization: The normalization of data to zero mean and unit variance before applying SLISEMAP is an important step. Could you elaborate on how this impacts both the interpretability and accuracy of your method, considering the varied nature of datasets in physical sciences?

Answer: The normalisation step is optional but very common in machine learning, including manifold visualisation methods such as PCA and t-SNE. The reason we use normalisation is the same as in other methods: to make different variables more comparable (especially with regularisation) and to increase numerical stability. However, the normalisation is reversible, since it is a linear transformation, if we want original units for the interpretation (an implicit example of this is in Section 4.1).

3. Detailed Performance Evaluation: The performance measures (permutation loss, local model stability, and neighbourhood stability) you used are intriguing. Can you offer more comprehensive insights into how these measures evaluate the effectiveness of SLISEMAP in visualizing and interpreting complex datasets?

Answer: As mentioned in the answer above, we have extended Section 3.3 with figures and discussions about what good and bad values imply. We also added more discussion of the results to Section 4.4.

4. Varied Field Applications: Your paper demonstrates the application of SLISEMAP across atmospheric science, high energy physics, and organic chemistry. Could you provide further details or additional examples illustrating how SLISEMAP's application and effectiveness vary across these fields?

Answer: As answered above, we have extended the comparison, analysis, and discussion in Sections 4.4 and 4.5.

5. Comparison with Existing Methods: Given the rise of supervised learning in physical sciences, how does SLISEMAP complement or enhance existing methods for interpreting complex models in these disciplines?

Answer: As mentioned in the answer above, SLISEMAP does not generate new, potentially unphysical, data and provides a more holistic view than just a single local explanation. We have added discussions to Sections 2, 3.2, and 3.3. We have also added a comparison to some local explanation methods in Section 4.4.

6. Handling Similarities Between Data Points: The introduction highlights a challenge in manifold visualization regarding the dependency of embedding on the measurement of similarities between data points. How does SLISEMAP address this issue, and what specific methods are employed to measure and interpret these similarities?

Answer: We have added the following sentences to the end of Section 2: "With SLISEMAP, we do not need to specify a distance metric in high-dimensional space. Rather, we measure how well the local model for one data item predicts the outcome of another data item." This tight coupling between the explanations and the embedding is what differentiates SLISEMAP from traditional manifold visualisation methods.

References:

[1] Anton Björklund, Jarmo Mäkelä, and Kai Puolamäki. “SLISEMAP: Supervised Dimensionality Reduction through Local Explanations”. In: Machine Learning 112.1 (2023), pp. 1–43. issn: 0885-6125, 1573-0565. doi: 10.1007/s10994-022-06261-1.

[2] Anton Björklund et al. “Explaining Any Black Box Model Using Real Data”. In: Frontiers in Computer Science 5 (Aug. 2023), p. 1143904. issn: 2624-9898. doi: 10.3389/fcomp.2023.1143904.

---

## [Editor Report · Decision Letter 1]

11 Jan 2024

Using Slisemap to interpret physical data

PONE-D-23-35222R1

Dear Dr. Seppäläinen,

We’re pleased to inform you that your manuscript has been judged scientifically suitable for publication and will be formally accepted for publication once it meets all outstanding technical requirements.

Kind regards,

Jayesh Soni

Academic Editor

PLOS ONE

Additional Editor Comments (optional):

The revisions have satisfactorily addressed the reviewers' comments, and the necessary changes have been implemented effectively. Before publication, please ensure that all figures and tables are correctly cited in the text, and the methodology is as clear as possible for the readers' comprehension.

---

## [Editor Report · Acceptance letter]

16 Jan 2024

PONE-D-23-35222R1 

PLOS ONE

Dear Dr. Seppäläinen, 

I'm pleased to inform you that your manuscript has been deemed suitable for publication in PLOS ONE. Congratulations! Your manuscript is now being handed over to our production team.

Kind regards, 

on behalf of

Dr. Jayesh Soni 

Academic Editor

PLOS ONE